# Standardised protocol for a prospective cross-sectional multicentre clinical utility evaluation of two dual point-of-care tests in non-clinical settings for the screening of HIV and syphilis in men who have sex with men

The ProSPeRo Network

**Correspondence to**
Laura Fernàndez-López;
lflopez@iconcologia.net

## ABSTRACT

**Introduction** Point-of-care dual tests (POCTs) for simultaneously detecting antibodies to HIV and syphilis (dual HIV-syphilis POCTs) have been developed recently and show encouraging performance compared with the reference tests in laboratory-based studies. As community-based voluntary, counselling and testing (CBVCT) services are effective providers of HIV and syphilis testing and counselling with high acceptability among men who have sex with men (MSM), the evaluation of the utility of these dual tests in CBVCT services is a high priority. This prospective cross-sectional study will conduct a clinical utility evaluation of two dual POCTs in non-clinical settings for the screening of HIV and syphilis in MSM. This master protocol outlines the overall research approach that will be used in four countries.

**Methods and analysis** MSM presenting at CBVCT services participating in the study for HIV/STI screening will be enrolled. The (WHO preapproved) dual POCTs to be evaluated will be SD Bioline HIV/Syphilis Duo (Abbot) and Dual Path Platform HIV-Syphilis Assay (Chembio). Trained staff will collect a capillary blood sample using finger prick blood to perform both POCTs according the manufacturers' instructions. An analysis of the feasibility of introducing the dual POCT for the screening of HIV and syphilis in MSM at CBVCT services will be performed, by assessing its acceptability and usability at CBVCT service among MSM users and providers.

**Ethics and dissemination** This core protocol was independently peer reviewed and approved by the Research Project Review Panel (RP2) of the WHO Department of Sexual and Reproductive Health and Research and by the WHO Ethics Review Committee (ERC). The protocol has been adapted to individual countries and approved by RP2, ERC and institutional review boards at each site. Results will be disseminated through peer-reviewed journals and relevant conferences.

## STRENGTHS AND LIMITATIONS OF THIS STUDY

⇒ To our knowledge, this is the first independent multicountry clinical utility evaluation of dual point-of-care tests (POCTs) for the screening of HIV/syphilis among men who have sex with men (MSM) in non-clinical settings.

⇒ This study will evaluate the feasibility of the introduction of dual HIV/syphilis test in community-based testing services, assessing the acceptability and the usability by users and providers of the services.

⇒ The study design uses a conceptual framework that considers different attributes working in an inter-related way to contribute to the feasibility of the introduction of dual HIV/syphilis POCTs in community-based voluntary, counselling and testing (CBVCT) services for the MSM screening, allowing a more accurate analysis of the feasibility.

⇒ Despite all the benefits of dual HIV/syphilis POCTs for MSM users of CBVCT services, it should be noted that treponemal antibodies persist after successful syphilis treatment, so additional confirmatory tests may be required to correctly identify active infections. The results of this study will reflect the attitudes of MSM users and providers of the participating CBVCT services and cannot be generalised to other CBVCT services and/or other populations.

## INTRODUCTION

HIV continues to be a major global public health issue with 1 700 000 people newly infected with HIV in 2019 and an estimated 38 million people living with HIV at the end of 2019.[1] In 2019, almost one quarter (23%) of global new adult HIV infections were among men who have sex with men (MSM). This population accounted for more than 40% of new infections in Asia, the Pacific and Latin America, and nearly two-thirds (64%) of new infections in western and central Europe and North America.[2] Also, worldwide syphilis is a highly prevalent infection among MSM. Since 2010 number of cases of syphilis have been increasing in developed countries, with

rates rising most rapidly among MSM.[3] In Europe, MSM are disproportionately affected by HIV and other STIs like syphilis, accounting for 39% of all new HIV diagnoses in 2019 and more than half (51%) of diagnoses where the route of transmission was known),[4] and for more than two-thirds (69%) of syphilis cases (with information on transmission category).[5]

In order to control the transmission of HIV and STIs and reduce their sequelae it is very important to provide screening or significantly enhanced testing of key populations and an accurate diagnoses in order to provide correct and early treatments.[6–8] Accurate, rapid and affordable point-of-care-tests (POCTs) could increase access to testing and identification of HIV and STIs in a single patient visit, including innovative delivery options, such us on-site delivery, community-based testing, as well as self-testing at home.[9]

A community-based voluntary counselling and testing (CBVCT) service is defined as any programme or service which offers voluntary HIV counselling and testing as one of its main activities, independently of clinical settings, targeted to specific groups of the population and clearly adapted and accessible to the communities to whom it is addressed.[10] The CBVCT services strengthen a comprehensive prevention strategy by increasing the number of engaged at-risk individuals who both become aware of their HIV and syphilis serostatus and by providing an entry point for care and treatment.[11–14] As described in the WHO consolidated guidelines on HIV testing services, community-based testing approaches may lead to earlier HIV and syphilis detection, as well as reaching people who are not routinely accessing health services, but are willing to test in a community-based HIV testing environment.[15]

Recently, dual tests that can be used at POC for simultaneously detecting antibodies to HIV and syphilis (dual HIV-syphilis POCTs) have been developed for use with finger-prick capillary whole blood specimens.[16] Some of these dual POCTs are now commercially available. To date, they have shown an encouraging performance compared with the reference tests in laboratory-based studies, but there is limited data on their utility in the field. As CBVCT services are effective providers of HIV and syphilis testing and counselling with high acceptability among MSM, evaluation of the utility of these dual tests in CBVCT services is a high priority.

The evaluation of these POCTs in a community setting is important as MSM at high risk of acquiring and transmitting STIs, including HIV, might face various barriers to accessing care and the CBVCTs are often their first entry point to the healthcare system. The use of POCTs in CBVCTs could therefore enhance the effectiveness of outreach screening in non-clinical settings because POCT results are rapidly available and reduce loss to follow-up and allow for timely counselling, referral and treatment. Syphilis can often be asymptomatic; undetected syphilis can result in serious long term complications and increased risk of HIV acquisition and transmission.

Screening and appropriate treatment for asymptomatic individuals infected with syphilis can reduce the risk of them developing serious long-term complication and interrupt onward transmission to their sexual partners. In the case of HIV early diagnosis of the infection is essential to ensure that patients are referred promptly for evaluation, provided with treatment and linked into counselling and related support services to help them reduce their risk for transmitting HIV to others.

There is a lack of independent evaluation of currently available POCTs (laboratory-based, clinic-based and utility evaluations), particularly in key populations and in low-income and middle-income settings.[9] Based on this, the Sexual and Reproductive Health and Research Department of the WHO has established the global ProSPeRo study (global Project on STI POCT). The overall objectives are to: (1) advise WHO Member States and other public health institutions on the performance characteristics of commercially available STI diagnostic tests that can be used at the POC; (2) assess the feasibility, acceptability of POCTs by both healthcare providers and clients/patients; and (3) support further implementation and roll-out of STI POCTs within national STI programmes by the provision of technical assistance tools.

ProSPeRo comprises three core components: (1) a laboratory-based arm assessing the performance characteristics of STI POCTs that have not yet been evaluated independently in the laboratory; (2) a clinical-based component to evaluate STIs POCT performance in the field compared with that of gold-standard laboratory tests among several STI high-risk and vulnerable populations worldwide and; (3) a clinical utility component assessing the feasibility and acceptability of STI POCTs among MSM in non-clinical settings in four countries within the WHO European region.

This master protocol refers to the third component of the ProSPeRo study, specifically to assess dual HIV/Syphilis POC technology in terms of its clinical utility.

Those clinical evaluation studies using this master protocol can adapt it to their local needs and can evaluate different dual HIV/syphilis POCTS.

### Objectives
The primary objectives of this utility evaluation are: (1) to assess the feasibility of introducing the dual POCT for the screening of HIV and syphilis in MSM at CBVCT services, by assessing its acceptability and usability among MSM users and providers of CBVCT services, and; (2) to assess the operational characteristics of the dual POCT for HIV and syphilis screening at the CBVCT services.

## METHODS AND ANALYSIS
### Study setting and design
This clinical utility evaluation is a multisite cross-sectional study of MSM presenting at CBVCT services for HIV/STI screening. The study will be implemented across multiple countries on the basis of locally adapted protocols. For

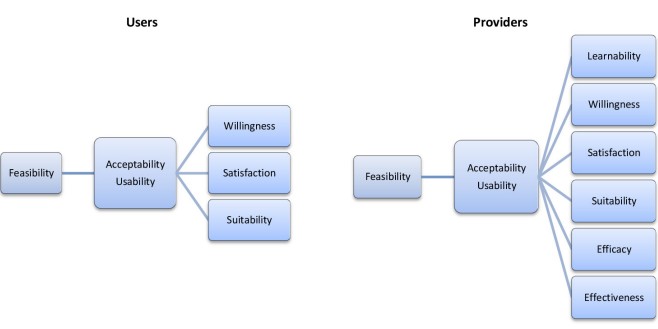

**Figure 1** Providers' and users' conceptual framework.

the purposes of this protocol, the term study site refers to an individual CBVCT service.

This paper is the master protocol and outlines the overall research approach which will be adapted accordingly for each site. Before implementation four CBVCT sites (from four different countries: Latvia, Slovenia, Spain and Ukraine) have been approved by the WHO in consultation with in-country researchers and providers, local authorities and WHO Country Offices (Latvia, Slovenia, Spain and Ukraine). Site selection criteria were based on: CBVCT service targeting MSM; access to a sufficiently large target population; ability to follow linkage to care within the local health services; staff capacity to perform the study in accordance with the study protocol; strong interest in working with new technologies; and offering testing for both HIV and syphilis as part of CBVCT services. A standardised site-assessment is implemented as part of the approval process for sites expressing an interest in participating. Site-specific protocols are developed with the WHO and the in-country principal investigator to agree and delineate the range of parameters and the minor changes needed to adapt the study to the local context while complying with this master protocol.

The global ProSPeRo study is ongoing with recruitment expected to be completed in all countries by late 2021.

## Study conceptual framework

The study conceptual framework has been designed following a model that explored the feasibility of the introduction of new health technology[17] (figure 1).

Regarding the CBVCT providers, the framework divides the concept of feasibility into two inter-related domains, acceptability and usability. Feasibility is defined as the process in which dual HIV/syphilis POCTs will be deployed by CBVCT providers leading to their acceptability and usability. These two domains have been further broken down into six subdomains: learnability, willingness, suitability, satisfaction, efficacy and effectiveness[18] (table 1). The operational characteristics that will be assessed and compared are also part of the conceptual framework: the clarity of kit instructions, the ease of use and interpretation of results are part of the learnability domain, while the waiting time for test results, the hands-on time and the training time required are part of the efficacy domain.

Regarding the CBVCT users (figure 1), the framework also divides the concept of feasibility into two inter-related domains, acceptability and usability, but these two domains are only broken down into three subdomains: willingness, suitability and satisfaction.

These attributes work in an interrelated way to contribute to the feasibility of the introduction of a new technology. Acceptability comprises positive perceptions, beliefs and attitudes towards dual HIV/syphilis POTCs among users and providers. Usability refers to the actions taken by the providers to apply the tool and its results to achieve specified outcomes, while usability among users refers to the actions they take to have the tests performed

| Table 1 | Acceptability and usability subdomain definitions |
|---|---|
| **Subdomains** | **Definition** |
| Learnability | Ability of the CBVCT providers to understand how to correctly perform the dual HIV/syphilis POTCs and accurately read the test results. |
| Willingness | CBVCT providers' intention to carry out a finger prick each time it is necessary, wait for the results, and refer the user when necessary. Regarding the CBVCT users, willingness has been defined as the intention to have the test performed on themselves, willingness to wait for test results, and if it is necessary, to follow the referral procedure. |
| Suitability | CBVCTs providers' beliefs that the test is relevant for their work and could be successfully integrated into existing services. Regarding CBVCT users, suitability has been defined as belief that the test is relevant in determining whether or not they have HIV and/or syphilis. |
| Satisfaction | CBVCT providers' feeling that the test is convenient to perform and that it is a process they like doing. Regarding the CBVCT users, satisfaction has been described as feeling that a test is convenient and that it is a process they would like to experience again. |
| Efficacy | CBVCT providers are able to make the effort and take the time to perform a test; read, interpret and record test results and also to refer the user if required, as part of their daily routine work. |
| Effectiveness | The enabling organisational and supporting systems, such as training, supervision, study aids, supplies, timers, storage and disposal are present or carried out and are integrated into existing routine protocols. |

CBVCT, community-based voluntary, counselling and testing; POCTs, point-of-care tests.

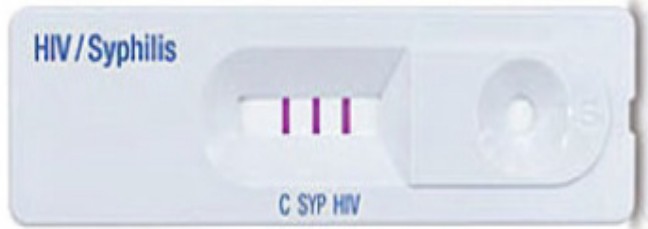

**Figure 2** Bioline point-of-care test.

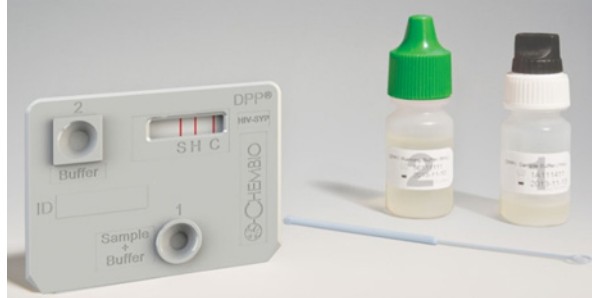

**Figure 4** DPP microreader, test device holder, DPP microreader with test device holder and test device. DPP, dual path platform.

on themselves believing that the test is accurate and convenient. In turn, if acceptability and usability are high among both providers and users, then implementation is feasible.

### Study participants
#### Inclusion criteria
The target population will be MSM. The term MSM will be used to describe those males who have sex with other males, regardless of whether or not they have sex with women or have a personal or social identity associated with that behaviour, such as being 'gay' or 'bisexual'. All participants have to be at least 18 years old to participate and sign a written consent.

CBVCT staff participating in the study will also be asked to complete a short questionnaire to evaluate the feasibility and operational characteristics of POCTs.

#### Exclusion criteria
MSM who will refuse to give consent, are younger than 18 years old, and/or have previously participated in the study.

### Description of the POCTs under evaluation
The tests to be evaluated will be SD Bioline HIV/Syphilis Duo (Abbott Diagnostics, USA; hereafter termed Bioline POCT) (figure 2) and Chembio Dual Path Platform (DPP) HIV-Syphilis Assay (Chembio, United States; hereafter termed Chembio POCT) (figure 3). Both will be single-use qualitative immunochromatographic assays for the simultaneous detection of antibodies against HIV types 1 and 2 (HIV 1/2) and/or *Treponema pallidum* (TP) in human serum, plasma, whole venous or finger pricked blood. In 2015, the Bioline POCT was accepted for the WHO list of prequalified in vitro diagnostics.[19]

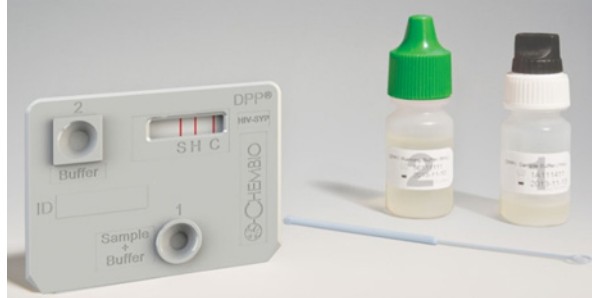

**Figure 3** Chembio point-of-care test kit. DPP, dual path platform.

Recently, the Chembio Company developed the DPP microreader (MR) to complete the Chembio DPP technology and minimise error due to subjective visual interpretation (figure 4). The DPP MR is a portable, battery-powered instrument that uses assay-specific algorithms to analyse the test and control line reflectance to determine the presence or absence of the antibodies to HIV and/or TP in the sample. The device is fitted to the Chembio POCT via a dedicated holder. The reader verifies the presence of the control line and measures colour intensity at each of the test line positions; it interprets the results using an algorithm including assay-specific cut-off values, and reports a positive, negative or invalid result.[20]

### Study procedure
#### Recruitment, enrolment and consent
For each site, clients will be recruited over 9 months (maximum) or until the required sample size is reached. Consecutive MSM presenting at the evaluation site or outreach settings will be informed about the study by the CBVCT provider performing the routine care (provider 1, see figure 5 patient flow chart). If the person is interested in participating (preconsent), another CBVCT provider (provider 2) will evaluate whether he fitted the inclusion criteria. If the potential participant fits the criteria and agrees to participate in the study, the latter CBVCT provider (provider 2) will take final consent and will perform the additional tests along with completion of the associated case report forms (CRFs). Users will be informed by the CBVCT provider and the consent form.

The participant recruitment part of the master protocol has been adapted to the specific testing procedures in the Slovenian CBVCT service. In this CBVCT service, the recruitment will be done when the client comes to the site for the second time to collect the results of laboratory tests. When the client arrives to collect the results, the receptionist will inform him about the study and if he is interested he will go with provider 2, who will explain the study in detail, obtain the informed consent and perform the dual tests. Provider 3 will do the second reading blind, and then both providers will pass the results to provider 1, who is the one who sees the results of laboratory tests, views all results together and informs the client of any reactive test result.

#### Specimen collection and result reading
Provider 1 will undertake a routine performance of standard tests according to local clinical procedures. If clients accept to participate, provider 2 will collect a capillary

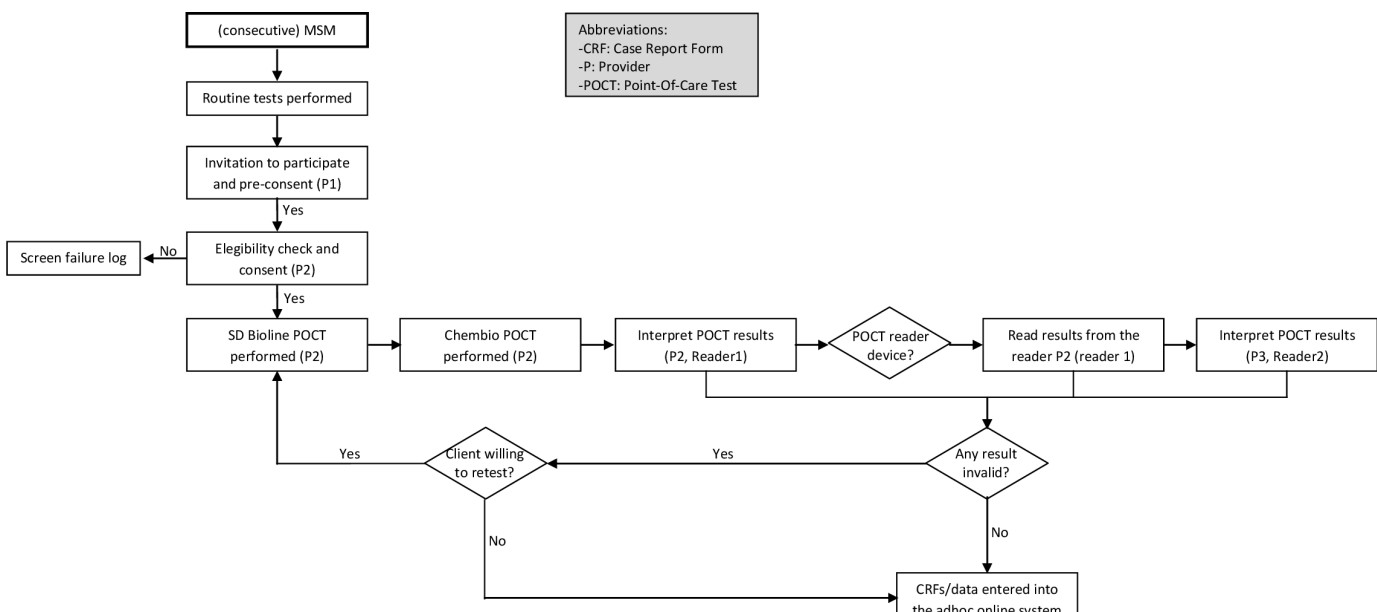

**Figure 5** Patient flow chart for a clinic-based evaluation of two dual HIV/syphilis POCTs. MSM, men who have sex with men; POCTs, point-of-care tests.

blood sample using finger prick blood to perform both POCTs according to the manufacturers' instructions; collect the required amount of capillary blood using the equipment provided in both test kits and wait the determined time (measured with a timer for each test) before reading the results. The finger is only pricked once: the first drop of blood will be used for the Bioline test and the second drop for the Chembio test. A double reader method (Reader 1-Reader 2 (R1-R2)) will be adopted for both tests to determine any variability in the interpretation of test results.[21] The MR (Chembio) will be read by R2 only (provider 3). R1 (provider 2) and R2 (provider 3) will be blind to each other's results and to the standard test results (read only by provider 1).

### Feasibility questionnaires
A user feasibility questionnaire (box 1) will be self-completed before and after the performance of the dual HIV/syphilis POCT and the routine tests and after the consent has been signed, but prior to receiving the tests results. A feasibility questionnaire (table 2) will be completed by each CBVCT provider who takes part in the study once the study period has finished, or when he/she leaves the study.

### Follow-up procedures
Follow-up and referral of the patients will be based on the results of the standard tests. Participants with a positive standard routine test result will be referred to the STI clinic or the reference hospital for confirmatory testing and treatment, following local guidelines. However, if the standard test result is negative, but one or both of the service providers' readings of the dual POCT(s) is positive for HIV and/or syphilis, the patient will be also referred for confirmation and treatment. Positive HIV POCT results will be preliminary and therefore must be

confirmed with the conventional screening test before the diagnosis of HIV infection is conclusively established. In the case of syphilis, the result will be considered as probable active syphilis; therefore referral will be made to the reference hospital for active infection confirmation.

### Outcomes
Primary outcome: Feasibility (assessed by the participant feasibility questionnaire and by the provider feasibility questionnaire). Secondary endpoints: Operational characteristics (assessed by the Operational characteristics of POC dual tests questionnaire); POCTs and routine tests results.

### Sample size
#### Sample size for tested individuals
The sample size calculation depends on the estimated proportion of people who have accepted to be tested by the dual POCTs for the screening of HIV and syphilis in a CBVCT service. As CBVCT services do not have such an estimate a proportion found in another study, of 81%, has been used.[22] Given an 81% population acceptance rate, 300 study subjects will be sufficient to estimate the feasibility of introducing the dual POCT for HIV and syphilis, with a 95% CI and a precision±5% units, and anticipating a replacement rate of 20% for those CBVCT service users who decline to participate.

The sample size of the master protocol will be adapted to the number of people routinely attended at the Baltic HIV Association. In this CBVCT centre, the sample size will be reduced to 150 study subjects.

#### Sample size for providers
It is expected than at least the 75% of the providers from the CBVTC service, who will receive the training and

Box 1    Feasibility questions for users and related subdomains

**User—feasibility questions**
Before the performance of point-of-care test (POCT)

**Willingness subdomain**
1. How long would you be willing to wait for the results of a dual test (*up to 20 minutes min, up to 30 min min, up to 1 hour hour, up to 2 hours hours, other, don't know*).
2. I would be willing to wait longer for the results of the dual test than for the separate tests (*strongly agree, agree, neither agree nor disagree, disagree, strongly disagree, don't know, don't want to answer*).
*After the performance of POCT*
3. Would you prefer two single tests or one dual test (*to check/test both infections at the same time)? (single, dual, it, don't know/don't care*).
3.1. If you prefer single test, why? (*don't want to be tested for HIV, don't want to be tested for syphilis, other*).

**Suitability subdomain**
4. I trust the results of the dual HIV/syphilis test (*strongly agree, agree, neither agree nor disagree, disagree, strongly disagree, don't know, don't want to answer*).
5. I believe the results of the dual HIV/syphilis test are more reliable than the tests performed routinely in this centre (two separate rapid tests for HIV and syphilis) (*strongly agree, agree, neither agree nor disagree, disagree, strongly disagree, don't know, don't want to answer*).

**Satisfaction subdomain**
6. I am more satisfied with the performance of the dual HIV/syphilis test than the separate tests for HIV and syphilis (*strongly agree, agree, neither agree nor disagree, disagree, strongly disagree, don't know, don't want to answer*).
7. In the future, I would prefer to use a dual HIV/syphilis test than two single tests to separately detect HIV and syphilis (*strongly agree, agree, neither agree nor disagree, disagree, strongly disagree, don't know, don't want to answer*).
8. I would recommend the dual HIV/syphilis test to a friend (*strongly agree, agree, neither agree nor disagree, disagree, strongly disagree, don't know, don't want to answer*).

perform the dual POCTs for the screening of HIV and syphilis, will answer the feasibility questionnaire.

### Project and data management

To ensure appropriate implementation of this master protocol, the following actions will be conducted: (1) development of site-specific study management plans including details of the roles and responsibilities of the study/evaluation team (the composition and number of study team members will be adapted at each site according to local need); (2) WHO monitoring visits and monitoring procedures to assess the progress and quality of the study at each evaluation site; (3) an internal (serum) and external (dried tube specimens) quality assurance process for ensuring accurate performance of the dual HIV/syphilis POCTs; and (4) a site-sensitive training programme for CBVCT staff in specimen collection and handling including performance and reading of the POCTs, as well as, familiarisation with the study standard operating procedures.

All data generated will be recorded using designed and piloted CRFs, which have been approved by the WHO. Paper versions will be stored securely at each study site as per local standard procedures. At regular intervals, data from these CRFs will be entered by a data manager at each site into a WHO provided secured laptop using an adhoc online system. Once data entry is completed, local data managers will be requested to check a random allocation of 10% of the data to reduce data entry error. Archiving (including destruction) of paper versions of the CRFs will be determined by the evaluation sites' own procedures. Only the data necessary to complete the project objectives will be included in the project database. Although the data will be stored on an IP secure website and processed by the study researchers, it belongs to each patient and they will be informed of how to request the deletion of the data at any time. The timeline for keeping data will be according to local and WHO policies.

### Data analysis

Subjects' demographic data, dual POCTs and routine tests results, follow-up of positives, data on knowledge and operational characteristics of dual POCTs will be summarised using descriptive statistics for aggregate and site level data.

For the feasibility analysis (first objective), data from feasibility questionnaires will be analysed in aggregates (taking into account the local practices when interpreting the results) and per centre (for those centres which had reached the expected number of recruited participants).

The questions in each subdomain will be likert items, most of them consisting of a discrete number of choices per question among the sequence: 'strongly disagree', 'disagree', 'no opinion', 'agree', 'strongly agree'. Some questions use other sequences of bipolar adjectives: 'very easy', 'quite easy', 'neither easy nor difficult', 'quite difficult', 'very difficult'.

Following the structure in the conceptual framework, the feasibility analysis will be performed in three stages (for individual questions, subdomains and domains): first calculate the median score for each question (excluding 'don't know/don't want to answer'), second the median score will be calculated for all questions within a subdomain, and lastly the total median score for all questions within a domain willbe reported.

In order to calculate the scores, a summated scores method will be used, calculating summated scores for each individual for each subdomain. The same weight will be considered for all the questions in each subdomain. Each total score will be divided by the number of items of the subdomain, obtaining a score ranging from 1 to 5 (from 1: highly in favour to 5: highly disagree). Scores will be calculated when all questions will be answered. For the qualitative interpretation of the score results, according the values assigned to the likert-type items, from 1 to 5 (1 being 'strongly agree', 2 'agree', 3 'no opinion', 4 'disagree' and 5 'strongly disagree'), the obtained domains' median scores, will indicate a high,

**Table 2** Feasibility questions for providers, operational characteristics and related subdomains

| Provider–feasibility questions | |
|---|---|
| **Learnability subdomain** | |
| 1. Overall, performing dual HIV/syphilis test is (Very easy, Quite easy, Neither easy nor difficult, Quite difficult, Very difficult, Don't know, Don't want to answer) | |
| 2. Correctly reading and interpreting the dual HIV/syphilis text result is (Very easy, Quite easy, Neither easy nor difficult, Quite difficult, Very difficult, Don't know, Don't want to answer) | |
| 3. Interpreting weak positive test result is (Very easy, Quite easy, Neither easy nor difficult, Quite difficult, Very difficult, Don't know, Don't want to answer) | |
| 4. The training offered was enough to perform the dual test (strongly agree, agree, neither agree nor disagree, disagree, strongly disagree, don't know, don't want to answer) | |
| **Willingness subdomain** | |
| 5. I am willing to perform the dual HIV/syphilis test instead of the separate HIV and syphilis tests in my CBVCT (strongly agree, agree, neither agree nor disagree, disagree, strongly disagree, don't know, don't want to answer) | |
| 6. Current supporting components of the study, including training, supervision and quality maintenance are sufficient to integrate the dual HIV/syphilis test into the routine activities in my CBVCT (strongly agree, agree, neither agree nor disagree, disagree, strongly disagree, don't know, don't want to answer) | |
| **Suitability subdomain** | |
| 7. I am confident in the results of the dual HIV/syphilis test (strongly agree, agree, neither agree nor disagree, disagree, strongly disagree, don't know, don't want to answer) | |
| 8. Routine dual HIV/syphilis testing should continue in my CBVCT service (strongly agree, agree, neither agree nor disagree, disagree, strongly disagree, don't know, don't want to answer) | |
| 9. Rapid dual HIV/syphilis tests could be successfully integrated in my CBVCT (strongly agree, agree, neither agree nor disagree, disagree, strongly disagree, don't know, don't want to answer) | |
| **Satisfaction subdomain** | |
| 10. In your opinion, how do new users feel about the dual HIV/syphilis tests? (Very positive, Quite positive, Neither negative nor positive, Quite negative, Very negative, Don't know, Don't want to answer) | |
| 11. Use of dual testing in this CBVCT reduces the workload (strongly agree, agree, neither agree nor disagree, disagree, strongly disagree, don't know, don't want to answer) | |
| 12. Dual testing is more acceptable to users than separate HIV and syphilis tests (strongly agree, agree, neither agree nor disagree, disagree, strongly disagree, don't know, don't want to answer) | |
| 13. Introducing dual HIV/syphilis tests will decrease user waiting time at the CBVCT service (strongly agree, agree, neither agree nor disagree, disagree, strongly disagree, don't know, don't want to answer) | |
| **Effectiveness subdomain** | |
| 14. The current supplier of HIV and syphilis tests will be able to provide the dual HIV/syphilis tests (strongly agree, agree, neither agree nor disagree, disagree, strongly disagree, don't know, don't want to answer) | |
| 15. Dual HIV/syphilis tests can be easily integrated into the national and/or regional HIV testing guidelines (strongly agree, agree, neither agree nor disagree, disagree, strongly disagree, don't know, don't want to answer) | |
| **Operational characteristics** | |
| 1. Clarity of kit instructions (difficult to follow, fairly clear, very clear, excellent) | Learnability subdomain |
| 2. Ease of use (complicated, fairly easy, very easy, excellent) | |
| 3. Ease of interpretation of results (difficult, fairly easy, very easy, unambiguous) | |
| 4. Rapidity of tests results (<20 min, 20–30 min, >30 min) | Efficacy subdomain |
| 5. Hands-on time (<5 min, 5 min, 10 min) | |
| 6. Training time required (<30 min, 30 min, 1 hour, >1 hour) | |
| 7. No of tests needed to be performed before being able to feel comfortable with POCT | |

CBVCT, community-based voluntary, counselling and testing; POCT, point-of-care test.

medium or low acceptability and usability. If acceptability and usability are high among both providers and users, then implementation is feasible.

For the second objective, data from routine tests, dual POCTs and confirmatory tests of all participant sites will be analysed in aggregates. Data regarding operational

tests characteristics from feasibility questionnaires will be also analysed in aggregates.

In order to validate the reading of the dual POCTs, the concordance between the two different readers will be estimated by calculating percentage of agreement (concordance) and kappa (κ for binary variables).

## Contextual survey

A contextual survey has been developed to be sent to the principal investigators of each participating CBVCT service in order to expand local contextual information about the participating CBVCT services. This will facilitate interpretation of data resulting from the study for each centre.

The questionnaire (online supplemental file 1) includes questions about: service characteristics and daily activities; procedures followed by the service regarding HIV and Syphilis testing (including confirmation and referral for those with a positive test result); research capacity and contextual information on testing, and some country sexual health indicators (laws regulations and/or policies related to age of consent for sexual health counselling and testing, prohibiting some sexual-related practices and sexual violence, supporting non-discrimination, criminalising or regulating sex work).

## Patient and public involvement

Patients, representatives of MSM communities and CBVCT service staff have been consulted during the development of this master protocol, specifically regarding participant recruitment and approach. Additional consultations have been held during adaptation of the master protocol to individual sites.

## Ethics and dissemination

This master protocol has been independently peer reviewed and approved by the Research Project Review Panel (RP2) of the WHO Department of Sexual and Reproductive Health and Research (SRH) and by the WHO Ethics Review Committee (ERC). It has also been adapted to individual countries and approved by RP2, ERC and institutional review boards at each site. Autonomy of the users to decide to participate in the study will be safeguarded by the division of the roles of taking preconsent on the one hand and performing the study on the other. The final consent will be taken by the CBVCT provider who performs the test, as he/she will also check if the user fits the inclusion criteria, for confidentiality reasons. Participation involves extracting two additional drops of blood from the fingertip to perform the new HIV and syphilis dual test in addition to the standard routine tests. The records concerning the participation will be used only for the purpose of the research project. Names will not be used on any study form or label on specimens or in any report resulting from the study. At the beginning of the study, a study identification number will be given and this number will be used on the forms and on the specimens.

## DISCUSSION

Implementation of dual POCT for HIV and syphilis in community-based services for MSM represents an opportunity to scale up integrated syphilis/HIV testing for this population. Although in several CBVCT services, single POCT for HIV and syphilis are already performed, implementation of dual POCT for both infections could increase syphilis testing for those only prone to test for HIV and vice versa in the case of syphilis.

Although there has been rapid development of new POCTs for STIs in recent years and there are some promising dual POCTs for HIV/syphilis in the pipeline and others already in the market, few of them have been well evaluated in a real-life setting. This has meant that there are still no formal WHO guidance and recommendations available on the implementation of these new tools for the diagnostics of HIV/STIs at the community level.

This paper describes the master protocol of the ProSPeRo study to conduct a clinical utility evaluation of dual POCTs for the screening of HIV and syphilis in MSM in non-clinical settings.

The results of this clinical utility evaluation, jointly with the results of the global ProSPeRo study will contribute to the advising of WHO member states and other public health institutions on the feasibility of dual POCTs for syphilis and HIV by both users and providers of CBVCT services. It will contribute to the evidence needed to develop the guidance for WHO member states on STI diagnostic tests that can be used at the POC and to support further implementation and rollout of those POCTs within national STI programmes.

**Acknowledgements** The authors are grateful to members of the WHO Research Project Review Panel (RP2) and the WHO Research Ethics Review Committee (WHO ERC) for their expertise and inputs regarding the master and site- specific protocols for this clinic-based evaluation. Our gratitude is extended to the global ProSPeRo network.

**Collaborators** ProSPeRo Network (Project on SexuallyTransmitted Infection Point-of-careTesting established by the Reproductive Health and Research Department of WHO): Italy: Massimo Mirandola; Latvia: Inga Upmace, Mara Vaselova; Slovenia: Mitja Ćosić, Simon Maljevac; Spain: Jordi Casabona, Juliana Reyes-Urueña, Laura Fernàndez-López, William Mejías, Ander Pazos; Ukraine: Andrii Chernyshev; United Kingdom: Rosanna Peeling; WHO: Ronald Ballard, Karel Blondeel, James Kiarie, Soe Soe Thwin, Igor Toskin.

**Contributors** The first draft of the manuscript was written by LF-L and JR-U. IT (chief and principal investigator) and RP conceived the whole ProSPeRo study. IT and JC conceived the clinical utility study and JR-U and LF-L developed the core study protocol, based on the other core study protocols of the ProSPeRo studies. The ProSPeRo network participated in the design of the study. IU, MV, MC, SM, WM, AP, AC, XX, MM, RB, KB, JK, SST, JC and IT led/will lead acquisition of data, contributed to adaptation of the master protocol, and commented on previous versions of the manuscript. All authors read and approved the final manuscript prior to submission.

**Funding** This work received funding from the UNDP-UNFPA-UNICEF-WHO-World Bank Special Programme of Research, Development and Research Training in Human Reproduction (HRP), a cosponsored programme executed by the World Health Organization (WHO) and Government of Canada.

**Disclaimer** Some of the authors are present or former staff members of the World Health Organization. The authors alone are responsible for the views expressed in this publication and they do not necessarily represent the views, decisions or policies of the institutions with which are affiliated.

**Competing interests** The POCT manufacturers disclose and furnish the WHO with the information and sufficient quantities of the product(s) free of charge in order to enable this evaluation as part of the WHO/RHR STI POC initiative. The WHO is entitled to evaluate and publish the trial results, and to exclusively control this evaluation and the content of the aforesaid publication. WHO shall submit any proposed publication to the manufacturers for review, comments received will be considered in good faith, but the decision to publish rests with the WHO.

**Patient and public involvement** Patients and/or the public were involved in the design, or conduct, or reporting, or dissemination plans of this research. Refer to the Methods section for further details.

**Patient consent for publication** Not applicable.

**Provenance and peer review** Not commissioned; externally peer reviewed.

**Open access** This is an open access article distributed in accordance with the Creative Commons Attribution 4.0 Unported (CC BY 4.0) license, which permits others to copy, redistribute, remix, transform and build upon this work for any purpose, provided the original work is properly cited, a link to the licence is given, and indication of whether changes were made. See: https://creativecommons.org/licenses/by/4.0/.

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
