## [Reviewer comments · BMJ Open]

ARTICLE DETAILS

TITLE (PROVISIONAL)	A standardized protocol for a prospective cross-sectional multi-centre clinical utility evaluation of two dual point-of-care tests in non-clinical settings for the screening of HIV and syphilis in men who have sex with men
AUTHORS	Fernández-López, Laura

VERSION 1 – REVIEW

REVIEWER	Lombard, Carl South Africa Medical Research Council, Biostatistics Unit
REVIEW RETURNED	13-Sep-2021

GENERAL COMMENTS	Abstract line 17. Here seven countries for the study are indicated whereas in line 51 of the paper it is stated as four. Thus some inconsistency exist. Sample size of 300 MSM per CBVCT site is indicated which is fine. The data analysis plan is however very vague on the analysis strategy. From the sample size it seems that a site specific analysis will be done but there is not indicated of how the study results will be used overall and whether a pooled analysis over all site will be done. For the later the clustering of the results at the site level have to be considered if confidence intervals and standard errors are calculated.
--

REVIEWER	Phang Romero Casas, Carmen Fundacao Oswaldo Cruz, Center for Health Technology in Health
REVIEW RETURNED	08-Oct-2021

GENERAL COMMENTS	The theme of this study is a very relevant issue for public health; major efforts must be made trying to improve the access to early diagnostic and treatment of HIV and syphilis in vulnerable groups like men who have sex with men (MSM), or pregnant women, which implies vertical mother-to-child-transmission with adverse events during pregnancy and deleterious consequences in newborns. This study protocol was already reviewed by Ethical committee, the ethical issues seems have been treated properly and is going to conduct a clinical utility evaluation of dual point-of-care tests (POCTs) for the screening of HIV and Syphilis in MSM in non-clinical settings. The manuscript is well written, methodology is enough detailed to be repeated and the analysis of results are clear. The questionnaires are also well structured and simple. I recommend the publication of the manuscript, even though just two comments deserve to be done. 1- Objectives (p.5): the second objective is to assess the operational
---

	characteristics of dual POCTs for HIV and Syphilis screening... the authors mentioned that “if possible” them will be compared with the operational characteristics of tests performed routinely. This is confused, the study will compare or will not?... if not, preferably I suggest to retire the phrase. Furthermore, the authors not more mention it along the entire manuscript. 2- In regards to the analysis, maybe is good to mention if the questions are going to be the same weight intra domain or between the domains to calculate and sum up the scores, and explain what the qualitative interpretation of the results will be.
--	--

VERSION 1 – AUTHOR RESPONSE

Reviewer: 1

Dr. Carl Lombard, South Africa Medical Research Council

Comments to the Author:

Abstract line 17. Here seven countries for the study are indicated whereas in line 51 of the paper it is stated as four. Thus some inconsistency exist.

We would like to thank to the reviewer for highlighting this mistake, we have changed from “seven” to “four” in the first paragraph of the abstract, accordingly to the reviewer's comment.

Sample size of 300 MSM per CBVCT site is indicated which is fine. The data analysis plan is however very vague on the analysis strategy. From the sample size it seems that a site specific analysis will be done but there is not indicated of how the study results will be used overall and whether a pooled analysis over all site will be done.

For the later the clustering of the results at the site level have to be considered if confidence intervals and standard errors are calculated.

Accordingly to the reviewer’s comment, the data analysis section has been extended, explaining better the analysis plan for the two objectives, including the plans for pooled analysis over all sites and site specific analysis.

The following paragraphs have been added:

- “Subjects’ demographic data, dual POCTs and routine tests results, follow-up of positives, data on knowledge and operational characteristics of dual POCTs will be summarized using descriptive statistics for aggregate and site level data.
- For the feasibility analysis (first objective), data from feasibility questionnaires will be analysed in aggregates (taking into account the local practices when interpreting the results) and per centre (for those centres which had reached the expected number of recruited participants).”

Reviewer: 2

Dr. Carmen Phang Romero Casas, Fundacao Oswaldo Cruz

Comments to the Author:

The theme of this study is a very relevant issue for public health; major efforts must be made trying to improve the access to early diagnostic and treatment of HIV and syphilis in vulnerable groups like men who have sex with men (MSM), or pregnant women, which implies vertical mother-to-child-transmission with adverse events during pregnancy and deleterious consequences in newborns.

This study protocol was already reviewed by Ethical committee, the ethical issues seems have been treated properly and is going to conduct a clinical utility evaluation of dual point-of-care tests (POCTs) for the screening of HIV and Syphilis in MSM in non- clinical settings.

The manuscript is well written, methodology is enough detailed to be repeated and the analysis of results are clear. The questionnaires are also well structured and simple.

I recommend the publication of the manuscript, even though just two comments deserve to be done.

1- Objectives (p.5): the second objective is to assess the operational characteristics of dual POCTs for HIV and Syphilis screening... the authors mentioned that “if possible” them will be compared with the operational characteristics of tests performed routinely. This is confused, the study will compare or will not?... if not, preferably I suggest to retire the phrase. Furthermore, the authors not more mention it along the entire manuscript.

Accordingly to the reviewer’s suggestion we have deleted the confusing sentence.

2- In regards to the analysis, maybe is good to mention if the questions are going to be the same weight intra domain or between the domains to calculate and sum up the scores, and explain what the qualitative interpretation of the results will be.

Accordingly to the reviewer's comment, the following sentence has been added to the paragraph explaining the calculation of scores in the data analysis section: "The same weight will be considered for all the questions in each sub-domain."

The following paragraph has been added into the data analysis section, in order to explain how will be the qualitative interpretation of the score results: "For the qualitative interpretation of the score results, according the values assigned to the likert-type items, from 1 to 5 (1 being "strongly agree", 2 "agree", 3 "no opinion", 4 "disagree" and 5 "strongly disagree"), the obtained domains' median scores, will indicate a high, medium or low acceptability and usability. If acceptability and usability are high among both providers and users, then implementation is feasible."

VERSION 2 – REVIEW

REVIEWER	Lombard, Carl South Africa Medical Research Council, Biostatistics Unit
REVIEW RETURNED	13-Dec-2021
GENERAL COMMENTS	Happy with the corrections and updates.
REVIEWER	Phang Romero Casas, Carmen Fundacao Oswaldo Cruz, Center for Health Technology in Health
REVIEW RETURNED	17-Dec-2021
GENERAL COMMENTS	Recommended for publication. I'm satisfied with modifications done by the authors following the suggestions from reviewer in order to improve the understanding of the manuscript.